**Data Availability Statement:** All relevant data are within the paper and its Supporting Information files.

# The three NADH dehydrogenases of *Pseudomonas aeruginosa*: Their roles in energy metabolism and links to virulence

Teri N. Hreha[1,2☯¤a], Sara Foreman[2,3☯], Ana Duran-Pinedo[4], Andrew R. Morris[5¤b], Patricia Diaz-Rodriguez[2,6¤c], J. Andrew Jones[2,7¤d], Kristina Ferrara[1,2], Anais Bourges[2], Lauren Rodriguez[2,7], Mattheos A. G. Koffas[1,2,7], Mariah Hahn[2,6], Alan R. Hauser[5], Blanca Barquera [1,2,3]*

1 Department of Biological Sciences, Rensselaer Polytechnic Institute, Troy, New York, United States of America, 2 Center for Biotechnology and Interdisciplinary Studies, Rensselaer Polytechnic Institute, Troy, New York, United States of America, 3 Department of Chemistry and Chemical Biology, Rensselaer Polytechnic Institute, Troy, New York, United States of America, 4 Department of Oral Biology, University of Florida, College of Dentistry, Gainesville, Florida, United States of America, 5 Department of Microbiology-Immunology, Feinberg School of Medicine, Northwestern University, Chicago, Illinois, United States of America, 6 Department of Biomedical Engineering, Rensselaer Polytechnic Institute, Troy, New York, United States of America, 7 Department of Chemical and Biological Engineering, Rensselaer Polytechnic Institute, Troy, New York, United States of America

☯ These authors contributed equally to this work.
¤a Current address: Department of Pediatrics, Washington University School of Medicine, St. Louis, MO, United States of America
¤b Current address: Abbott, Abbott Park, IL, United States of America
¤c Current address: Department of Chemical Engineering and Pharmaceutical Technology, Universidad de La Laguna, Tenerife, Spain
¤d Current address: Department of Chemical, Paper and Biomedical Engineering, Miami University, Oxford, OH, United States of America
* barqub@rpi.edu

## Abstract

*Pseudomonas aeruginosa* is a ubiquitous opportunistic pathogen which relies on a highly adaptable metabolism to achieve broad pathogenesis. In one example of this flexibility, to catalyze the NADH:quinone oxidoreductase step of the respiratory chain, *P. aeruginosa* has three different enzymes: NUO, NQR and NDH2, all of which carry out the same redox function but have different energy conservation and ion transport properties. In order to better understand the roles of these enzymes, we constructed two series of mutants: (i) three single deletion mutants, each of which lacks one NADH dehydrogenase and (ii) three double deletion mutants, each of which retains only one of the three enzymes. All of the mutants grew approximately as well as wild type, when tested in rich and minimal medium and in a range of pH and [Na$^+$] conditions, except that the strain with only NUO ($\Delta nqrF\Delta ndh$) has an extended lag phase. During exponential phase, the NADH dehydrogenases contribute to total wild-type activity in the following order: NQR > NDH2 > NUO. Some mutants, including the strain without NQR ($\Delta nqrF$) had increased biofilm formation, pyocyanin production, and killed more efficiently in both macrophage and mouse infection models. Consistent with this, $\Delta nqrF$ showed increased transcription of genes involved in pyocyanin production.

**Funding:** This work was supported by a grant from the National Science Foundation. Grant Number: 1616674 to BB. The funders had no role in study design, data collection and analysis, decision to publish, or preparation of the manuscript.

**Competing interests:** The authors have declared that no competing interests exist.

## Introduction

*Pseudomonas aeruginosa* is an opportunistic pathogen characterized by an almost ubiquitous environmental presence and a broad infectious profile [1,2]. This microbe can inhabit a wide range of distinct niches and give rise to a variety of chronic infections in human hosts, many of which are life-threatening [3]. *P. aeruginosa* is the 4th most common nosocomial pathogen and the primary cause of mortality in cystic fibrosis (CF) patients [4,5]. The ability of this microbe to proliferate in so many different, and frequently hostile, environments has been attributed to its robust adaptability, arising from the flexibility of its metabolic processes [6–8].

A key point where this metabolic flexibility is apparent is in the organization of the respiratory chain, which is one of the most highly branched among bacteria. The respiratory chain of *P. aeruginosa* includes several dehydrogenases, the cytochrome $bc_1$ complex, and five terminal oxidases that operate at different concentrations of oxygen, as well as alternative electron acceptors that operate under anaerobic conditions[9]. It has been shown that this bacterium selectively expresses different respiratory enzymes/pathways depending on the availability of nutrients, oxygen, and other electron acceptors [10–14]. Such adaptations are important for colonization of infection sites, particularly in the lungs of patients suffering CF, where the bacteria are challenged with low nutrient and oxygen availability [5]. Therefore, this respiratory flexibility is likely to be a key factor in the success of *P. aeruginosa* as an opportunistic pathogen.

Perinbam et. al. recently identified a distinct shift in NADH metabolism which is associated with virulence in *P. aeruginosa* [15]. At the beginning of its respiratory chain, *P. aeruginosa* has three different NADH dehydrogenases which are responsible for the oxidation of cellular NADH: NUO, NQR, and NDH2 [9,16]. All three of these enzymes carry out the same redox reaction, accepting electrons from NADH and passing them to the quinone pool, but they differ in their ion pumping and energy conservation properties. NQR and NUO conserve energy by coupling the electron transfer to the translocation of ions across the cell membrane, contributing to an electrochemical membrane gradient [17–20]. NDH2 catalyzes the same redox reaction without ion pumping or energy conservation, but its activity has been implicated in a rescue-redox system than can balance the NADH/NAD$^+$ ratio to avoid toxic effects of excess NADH [17,21]. Given the highly branched character of respiratory pathways in *P. aeruginosa*, it is likely that the three NADH dehydrogenases make the organism more adaptable. However, little is known about their actual physiological roles.

Two previous publications have addressed these questions. Torres et. al. characterized a series of deletion mutants and concluded that NUO and NDH2 together are the primary NADH dehydrogenases during aerobic growth in rich medium (LB), while NUO is required for anaerobic growth and virulence in plant (lettuce) and insect (*Galleria mellonella*) models [22]. They also concluded that NQR has minimal activity and plays only a minor role in *P. aeruginosa* physiology. In contrast, Liang et. al. reported that NQR is the most active NADH dehydrogenase in wild type *P. aeruginosa* (PAO1) during aerobic growth in rich medium [23]. These inconsistencies underline the importance of further study of the roles of these enzymes.

Here, we have characterized the roles of the three NADH dehydrogenases using a series of single-deletion mutants, each of which lacks one of the enzymes, and a series of double-deletion mutants, each of which has only one of the three. All of the mutant strains were able to grow well in both rich and minimal medium, although the strain lacking NUO consistently showed an extended lag phase prior to entering exponential growth. In each of the mutants, the enzymes that remain all contributed to NADH dehydrogenase activity, in both exponential and stationary phases. From this we concluded that *P. aeruginosa* does not switch between different NADH dehydrogenases in different growth conditions. Instead, the presence of three

parallel enzymes confers resilience on its energy production systems. Surprisingly, we also discovered that, in some of the deletion mutants, the virulence factor pyocyanin, which is normally characteristic of stationary phase, began to be produced earlier, and in much larger quantities than in the wild type. We tested one such strain, the one lacking NQR, in two model host systems, macrophages and mice, and found that in both cases the mutant bacteria had become much more effective in killing the mammalian host cells. These findings suggest that in *P. aeruginosa*, NADH metabolism is closely involved in the control of virulence.

## Materials and methods

### Bacterial strains and growth conditions

The wild-type PAO1 and single mutant strains were purchased from the Two-Allele Library [24]. All strains used for this study are shown in Table 1. For clarity, the single deletion mutants are designated with the 'Δ' symbol, while the transposonal insertion mutant nomenclature is indicated in the strain descriptions provided in Table 1. Strains were grown in Luria Bertani (LB) broth (Miller) unless otherwise specified. When used, antibiotic concentrations

**Table 1. Strains and Plasmids used in this study.**

| Strain | Description | Source |
|---|---|---|
| *Pseudomonas aeruginosa* | | |
| PAO1 | Wild type | [62] |
| Δ*nqrF* | PW6010, *nqr*F-G08:: ISlacZ/hah (*nqrF*::Tn) | [24] |
| Δ*nuoG* | PW5420, *nuo*G-C11:: ISphoA/hah (*nuoG*::Tn) | [24] |
| Δ*ndh* | PW8644, *ndh*-F12:: ISlacZ/hah (*ndh*::Tn) | [24] |
| *nqr* pHERD28C-his-NQR | *nqr* operon cloned into the c pHERD28C-his-NQR vector | This study |
| Δ*nqrF*Δ*nuoG* | Chromosomal deletion of *nqrF* in the Δ*nuoG* background | This study |
| Δ*nqrF*Δ*ndh* | Chromosomal deletion of *nqrF* in the Δ*ndh* background | This study |
| Δ*nuoG*Δ*ndh* | Chromosomal deletion of *ndh* in the Δ*nuoG* background | This study |
| *Escherichia coli* | | |
| S17 lpir | *thi pro hsdR hsdM recA* RP4 2-Tc::Mu-KnR::Tn7 (TpR, SpR, SmR) | This study |
| NEB 5-a | *fhuA2 Δ(argF-lacZ)U169 phoA glnV44 Φ80 Δ(lacZ)M15 gyrA96 recA1 relA1 endA1 thi-1 hsdR17* | NEB |
| Plasmids | | |
| pEX18Gm | *sacB* counter-selectable suicide vector GmR | This study |
| pEX18Gm_BbsI | *sacB* counter-selectable suicide vector GmR | This study |
| pEX18Gm-*nqrF*KO | pEX18Gm vector containing the *nqrF* gene deletion assembly | This study |
| pEX18Gm-*ndh*KO | pEX18Gm vector containing the *ndh* gene deletion assembly | This study |
| pHERD28T-His | 6X-histidine- araC-pBAD, TpR | [25] |
| pHERD28T-His-NQR | *nqr* operon cloned into the pHERD28T-His | This study |
| pHERD28C-His-NQR | pHERD28T-his-NQR CmR replacing TmR | This study |
| pKD3 | Source of the chloramphenicol resistant cassette (CmR) | [26] |

were as follows: *Escherichia coli* strains were grown in 30 μg/mL kanamycin, 10 μg/mL tetracycline, 25 μg/mL gentamicin, and 12 μg/mL chloramphenicol. *P. aeruginosa* strains were grown in 50 μg/mL tetracycline, 100 μg/mL chloramphenicol, 100 μg/mL gentamicin. The NQR complementation strain pHERD28T-Cm-His-NQR-Cm was grown in the presence of 12 μg/mL chloramphenicol and 0.2% (w/v) arabinose as an inducer of expression of the *nqr* operon.

## DNA manipulations and mutant strain construction

Double deletion mutants were constructed from the single mutant strains via a two-step allelic exchange protocol adapted from Hmelo et. al. [27] (Tables 1 and 2). Single and double deletion mutants were verified by PCR and DNA sequencing.

## NQR complementation

The *nqr* operon was cloned into the pHERD28T-Cm-His-NQR-Cm using standard molecular biology protocols (Tables 1 and 2). The chloramphenicol resistance marker was amplified from pKD320 and inserted into the pHERD28T-HIS-NQR. pHERD28T-HIS-NQR backbone was amplified using Herculase Fusion polymerase. Insert and backbone were purified using EZNA Cycle Pure Kit (Omega BioTek) and assembled using the Gibson Assembly kit (NEB). The resulting mixture was transformed into Chemically Competent DH5alpha cells and

**Table 2. Primers used in this study.**

| Primer | Sequence 5' to3' | Description/used for |
|---|---|---|
| nqrF_cfrm_R1 | cgtgatcggattcgagattt | Amplification of the *nqrF*- IS*lacZ*/hah 590bp sequence for confirmation of transposon insertion in Δ*nqrF* mutant |
| lacZ_cfrm | gcgtagatacgacgcacca | |
| ndh_cfrm_R1 | ctacccatcagattgcccat | For amplification of *ndh*- IS*lacZ*/hah sequence with above LacZ F primer for confirmation of transposon insertion in Δ*ndh* mutant |
| nuoG_cfrm_F1 | tatccacgtagacggcaaga | Amplification of *nuoG*-IS*phoA*/hah sequence for confirmation of transposon insertion and location in Δ*nuoG* mutant |
| hah138_cfrm | cgggtgcagtaatatcgccct | |
| pEX18_BbsI_St_F | agcttgcaggtcttcgacggaagacctcgact | Generation of the sequence containing two BbsI restriction sites to be inserted into the pEX18 vector to create pEX18Gm_BbsI |
| pEX18_BbsI_St_R | ctagagtcgaggtcttccgtcgaagacctgca | |
| PaNqrKO_HR_Down_F | gggcccgaagactagaccagcgcgagaatatcctgctgga | Amplification of the downstream region of the *nqr* operon |
| PaNqrKO_HR_Down_R | gggcccgaagactatcgcgaaatagttgcggtaatcgcc | |
| PaNqrKO_HR_UP_F | gggcccgaagactattgccgtagtactcaccgcggcattg | Amplification of the upstream region of the *nqr* operon |
| PaNqrKO_HR_UP_R | gggcccgaagactaggtccaggccacgttttatcttg | |
| F_NqrKO_Scrn | tggcgatcatcatgttctcc | Amplification of sequence -1049 bp to +1022 bp of the *nqr* operon |
| R_NqrKO_Scrn | gtcatggacgttctccttgg | |
| ndh_Down-HR-F1 | gggcccgaagactagacccgagccacgcctcaagctgc | Amplification of the downstream region of the *ndh* gene |
| ndh_Down-HR-R1 | gggcccgaagactagtcgtattgctggccagcctcatgc | |
| ndh_Up2-HR-F1 | gggcccgaagactattgcatcaacctgctcagcctgaagg | Amplification of the upstream region (1) of the *ndh* gene |
| ndh_UP1-HR-R1 | gggcccgaagactacctcggggctcagtcggctggaca | |
| ndh_Up2-HR-F2 | gggcccgaagactagaggaccagcagaaggtcgagcag | Amplification of the upstream region (2) of the *ndh* gene |
| ndh_Up2-HR-R2 | gggcccgaagactaggtccgacgatcacgatgcgatgg | |
| UP_Ndh2_Seq_F | cctggaaaagcacatcgaccac | Amplification of sequence -395 bp to +198 bp of the *ndh* gene |
| Ndh2_upGene_R | cgcttcatcaatctcgtcgacg | |
| Cm_FWDw/Homology | cttttctgtgactggtgagtttctcatcgcagtactgttgtattc | Amplification of the chloramphenicol resistance cassette from the pKD3 plasmid |
| Cm_REVw/Homology | gattcacaagaaggattcgacatggagaaaaaaatcactggatatacc | |
| pHERD28His_FWDNoTm | catgtcgaatccttcttgtgaatc | Amplification of the pHERD28T-HIS-NQR backbone |
| pHERD28His_REVNoTm | gaaactcaccagtcacagaaaag | |

screened via restriction digest. pHERD28C-His-NQR-Cm from DH5alpha was transformed into wild type (PAO1) by electroporation, according to Choi et al. [28]. 6 mL of an overnight culture was centrifuged, washed twice with 1mL aliquots of 300 mM sucrose, and resuspended in 300 μL 300 mM sucrose. 300 ng of vector were mixed with 100 μL of competent cells, and electroporated (20 μF, 200 Ohms, and 2.5 kV). Immediately after the voltage was applied, 1 mL of LB broth was added to the cells, and they were allowed to recover for 1 hour at 37°C, before plating on LB plus appropriate antibiotics. The pHERD28T-HIS-NQR strains was grown in LB in the presence of 0.2% (w/v) arabinose to induce the expression of the *nqr* operon. The expression of NQR was tested by a Western blotting using anti-Histidine tag antibodies. The NQR complex was partially purified using a Ni-NTA column using a similar protocol as reported before [29].

## Growth curves

The growth of the different strains was followed in similar way as reported earlier [30]. Medium containing 10 g/L tryptone, 5 g/L yeast extract and 60 mM bis tris propane was prepared and adjusted to feature the desired pH and [$Na^+$] conditions to be tested. Minimal medium adapted from synthetic CF sputum medium (SCFM) [31] was prepared containing 0.2 M $NaHPO_4$, 0.2 M $Na_2HPO_4$, 2.28 mM $NH_4Cl$, 14.9 mM KCl, 10 mM MOPS, 271 μM $K_2SO_4$, 2.5 g/L casamino acids, 1.754 mM $CaCl_2$, 0.606 mM $MgCl_2$, 36 μM $FeSO_4$, 3 mM glucose and the desired concentration of NaCl.

Overnight cultures of WT and mutant strains were grown in 5mL LB (Miller) for 16 hours at 37°C with shaking at 200 rpm. $OD_{500}$ was measured and the cultures were diluted to an $OD_{500}$ of 1.0 using fresh LB broth of the corresponding pH and minimum [$Na^+$] to be tested. 10 μL of the diluted culture was used to inoculate 190 μL of fresh medium in a 96-well plate (Costar transparent flat bottom) to a starting $OD_{500}$ of 0.05. Plates were covered and incubated in at 37°C with continuous orbital shaking at 217 rpm (Tecan Infinite Magellan M1000 Pro). $OD_{500}$ was measured every 30 minutes over 20 hours of growth. $OD_{500}$ values were corrected for background absorbance and averaged across two biological replicates with three technical replicates each, with standard deviation calculated accordingly. Doubling times and the related statistical analyses were calculated in R using the growth-rates package with the easy linear fitting method [32, 33].

## Membrane preparation and enzymatic activity assay

Cultures in LB were inoculated from overnight cultures (15 h) to a starting $OD_{600}$ of 0.05 and grown aerobically until mid-exponential (6.5 hours) and stationary (22 h) phases. Cells were harvested by centrifugation in a Sorvall SLC-6000 rotor for 30 min at 3860 x g at 4°C and washed with TCDG buffer containing 10mM Tris-HCl, 140 mM choline chloride, 10% (v/v) glycerol and 0.5 mM dithiothreitol, pH 7.5. Cells were lysed via three passes through a microfluidizer cell disrupter at ~80 psi in the presence of PMSF and DNaseI. Cell debris was removed from solution by centrifugation at 5856.4 X g, 30 min, at 4°C and the remaining supernatant was centrifuged for at least 5 h in a Beckman Type Ti45 rotor at 185511.4 x g to collect the cell membranes. Membranes were resuspended in a small volume of TCDG buffer and frozen at -80°C until needed. Membrane protein concentration was determined using the Pierce Rapid Gold BCA Protein Assay Kit (Thermo). Measurements were done in triplicate.

NADH:Ubiquinone oxidoreductase activity was measured spectrophotometrically following the changes in absorbance at 340 nm ($\varepsilon_{NADH}$ = 6.22 $mM^{-1}$ $cm^{-1}$) as reported previously [34]. Assays were conducted in 1mL reaction volumes containing 100 mM NaCl, 50 μM ubiquinone-1 (UQ1), and 25 μg/mL membrane protein. Reactions were initiated by the addition

of 100 μM NADH, or 100 μM deamino-NADH (dNADH) where specified, and substrate absorbance was measured for 50 seconds following NADH addition.

## RNA-Seq

Wild type and Δ*nqrF* strains were inoculated from overnight cultures (15 h) to a starting $OD_{600}$ of 0.02 in 50 mL LB medium containing 50 mM NaCl. Three cultures of each strain were grown aerobically at 37˚C with shaking at 200rpm until mid-exponential (4 h) and stationary (15 h) phases. $1.5 \times 10^8$ cells were harvested by centrifugation at 16,000 x g for 1 min at 4˚C and rinsed with 1mL of ice-cold phosphate buffered saline (PBS). The three pellets for each strain and growth phase were frozen with "RNA-later" (ThermoFisher). RNA extraction and processing, library preparation, and Illumina sequencing were performed by GENEWIZ (South Plainfield, NJ, USA). GENEWIZ processed the resulting data as follows: reads were evaluated for sequence quality and trimmed using Trimmomatic [35]. Trimmed reads were mapped to the *P. aeruginosa* reference genome (https://bacteria.ensembl.org/Pseudomonas_aeruginosa_pao1/Info/Index/) using the Bowtie2 aligner and unique gene hit counts were calculated using featureCounts from the Subread package [36,37]. Differential gene expression analysis was carried out using DESeq2 [38]. The Wald test was used to generate log2-fold changes and P-values.

## Pyocyanin assay

Pyocyanin was extracted using a modified version of the protocol described by Koley et al [39]. 5 mL cell cultures were inoculated with $1 \times 10^8$ cells from an overnight culture and allowed to grow at 37˚C in an orbital shaker operating at 200 rpm for the desired length of time. Following growth, cells were removed by centrifugation and the cell supernatant was subjected to an organic extraction using 1–3 mL of chloroform. The organic lower layer was transferred to a new tube and centrifuged for 1 min at maximum speed. The organic supernatant was transferred to a new tube and dried under a $N_2$ stream until no solvent remained. The resulting pellet was resuspended in 1 mL 50 mM Tris-HCl pH 8.0, and absorbance was measured at 690 nm. The concentration of pyocyanin was determined using an extinction coefficient of 4,130 $M^{-1} cm^{-1}$ [39]. For pyocyanin production in the complementation strain, concentration was normalized by the $OD_{600}$ of the culture at the time of harvesting.

## Biofilm quantification

Biofilm formation was assessed according to the method outlined by Tram et. al. [40]. Overnight cultures were diluted to an $OD_{600}$ of 0.5 and transferred in 100 μL aliquots to a 96-well plate (Nunc). The plates were incubated at 37˚C, without shaking, for 6 hours (mid attachment) or 24 hours (mature biofilm) [40,41]. Following incubation, the supernatant was removed from each well, and the attached biofilm was washed three times with water, then incubated in 20μL 1% (w/v) crystal violet (Fisher) for 15 minutes. Wells were then washed three times with water, and the remaining crystal violet was extracted from the biofilm with 100 μL of MBDS (modified biofilm dissolving solution) consisting of 10% (w/v) SDS and 80% EtOH. The resulting MBDS solution was transferred to a new 96-well plate and absorbance at 600 nm was read in a plate reader (Tecan Infinite Magellan M1000 Pro).

For image analysis, overnight cultures were diluted to an $OD_{600}$ of 0.5 and transferred in 200 μL aliquots to an 8-well coverslip. Samples were incubated at 37˚C for 6 or 24 hours. Biofilms were washed twice with phosphate-buffered saline (PBS) and fixed with 4% (v/v) paraformaldehyde in PBS. Fixed biofilms were permeabilized with 1% (v/v) TritonX-100 in PBS, and then stained with 30 μM propidium iodide. The stained biofilms were washed 5 times with

PBS before imaging. Images were captured by a Nikon eclipse Ti-U inverted microscope, equipped with a scan module, using a Nikon Plan Flour 60X, 1.30 oil DIC objective lens, and an infrared pulsed laser at 970 nm with a 530/43 nm emission filter. Z-stacked images were collected over 10 μm, and the image size was 20 μm × 20 μm (256 pixels × 256 pixels). Images were analyzed using VistaVision software (ISS, Colorado Springs, CO). Biofilm analysis was done through COMSTAT2 (www.comstat.uk) [42,43].

## Antibiotic resistance determination

Antibiotic resistance was determined by inoculating 4 μL of an exponential phase culture ($OD_{600}$ = ~0.5) into 1 mL of LB with appropriate antibiotic in a 24-well plate. Plates were grown with shaking at 37˚C. The MIC was determined to be the concentration at which no detectable growth could be seen 24 hours after inoculation.

## Macrophage toxicity model

Overnight cultures of *P. aeruginosa* WT and mutants were grown from a single colony in Dulbecco's modified Eagle's medium (DMEM) (Gibco), 10% (w/v) fetal bovine serum (FBS) (Hyclone) and 36 μM $FeSO_4$. Overnight cultures were diluted 1:100 in 10 mL DMEM + 10% FBS and grown for approximately three hours before 250,000 cells in a total of 200 μL were added to the macrophages. RAW 264.7 cells from a murine macrophage cell line (ATCC, Manassas, Virginia) were maintained in DMEM (Gibco) supplemented with 10% (w/v) FBS. Macrophages were plated at a density of 12,500 cells/cm$^2$ on 24-well tissue culture polystyrene plates. After that, cell monolayers were incubated for 6 hours at 37˚C and 5% $CO_2$ with bacterial suspensions using a multiplicity of infection (MOI) of 1. Additionally, the initial number of cells was assessed by DNA measurement via PicoGreen assay (Invitrogen) per manufacturer's instructions. Calf thymus DNA (Sigma) was used as standard.

After the 6-hour incubation, the supernatant was collected, and macrophages were washed with DPBS (Lonza) and fixed with formalin 10% (Fisher) overnight at 4˚C. The next day wells were washed twice with DPBS and were imaged using a Zeiss Axiovert microscope. Cell toxicity was evaluated using the Cytotoxicity Detection KitPLUS (lactate dehydrogenase, LDH; Roche). Supernatants were centrifuged at 2,500 rpm for 5 min at 4˚C and assayed in duplicate following manufacturer instructions. Absorbance was measured after 10, 20 and 25 minutes of reaction at 490 nm using a plate reader (Biotek).

## Mouse infection model of acute pneumonia

*P. aeruginosa* cultures were grown overnight in 5 mL of MINS medium [44] at 37˚C with shaking (250 rpm), and then sub-cultured 1:100 into fresh MINS and regrown to exponential phase. The bacteria were collected by centrifugation and resuspended in PBS. Six- to eight-week-old female BALB/c were anesthetized by intraperitoneal injection of a mixture of ketamine (100 mg/mL) and xylazine (20 mg/mL). Mice were infected intranasally with the indicated colony-forming units (CFU) of bacteria in 50 mL of PBS. Bacterial inocula were confirmed by plating of serial dilutions onto Vogel-Bonner minimal (VBM) medium agar. All experiments were approved by the Northwestern University Institutional Animal Care and Use Committee.

For determination of the bacterial numbers in the lungs, mice were infected intranasally with ~5–6 x 10$^5$ CFU of bacteria. All mice were euthanized at 24 hours post infection and the lungs were aseptically removed and homogenized in 5 mL of PBS. The bacterial load was determined following plating of serial dilutions onto VBM agar and incubation at 37˚C for 24 hours. The results are expressed as the ratio of CFU recovered per lung (output) to the CFU

present in the initial inoculum (input). The data shown are grouped from three independent experiments (n = 15 mice per strain). The black line indicates the geometric mean for each group.

For survival experiments, mice were infected intranasally with ~4 x 10$^6$ CFU of the indicated bacterial strain. In all experiments, the mice were sacrificed when severe illness developed and were scored as dead. Severe illness was pre-defined as decreased activity, ruffled fur, weight loss of > 5% baseline body weight, or altered respiratory rate <90 or >170. Mice were monitored every 8 hours for evidence of severe illness over the first 3 days post-infection, then twice per day on the fourth day, since severe illness usually evolved during the first 72 hours. During monitoring, mice were immediately (within 30 minutes) euthanized if noted to meet criteria for severe illness. Monitoring staff were trained to recognize the criteria used to define severe illness and in the techniques of euthanasia. Mice were euthanized by cervical dislocation or bilateral thoracotomy while under ketamine/xylazine anesthesia or following carbon dioxide inhalation or injection with Euthasol. These experiments were performed in 2013, so information on the number of mice that died prior to meeting criteria for euthanasia is not available. Survival was monitored for 96 hours after infection. This experiment was repeated twice, and the results shown are from a single experiment (n = 10 total mice per group). All of these mice were eventually euthanized during the course of the experiment because they met criteria for severe pneumonia. In an effort to avoid influencing the progression of the infections, resuscitation treatments and analgesics were not administered. An exponential-rank test was used to analyze differences in mouse survival. A p-value < 0.05 was considered significant.

## Results

### NADH dehydrogenase deletion strains

To investigate the roles of the three NADH dehydrogenases of *P. aeruginosa*, we used mutant strains, each lacking one or more of these enzymes. We obtained three single-deletion strains (Δ*nqrF*, Δ*nuoG*, Δ*ndh*) from the University of Washington transposon library [24] and constructed three double-deletion strains, each of which retains only one of the three NADH dehydrogenases: NQR (Δ*nuoG*Δ*ndh*), NUO (Δ*nqrF*Δ*ndh)*, and NDH2 (Δ*nqrF*Δ*nuoG)*.

### Growth properties

We examined the effect of the missing enzyme(s) by comparing the growth of each mutant with wild type in liquid cultures. We tested growth in a rich medium (Luria Bertani, LB) and "Synthetic Cystic Fibrosis Sputum Medium" (SCFM) a minimal medium designed to mimic the chemical conditions in the cystic fibrosis lung [31], which can provide insights into the growth of these strains in infection-relevant settings. Since some of the NADH dehydrogenases conserve energy by pumping cations across the cell membrane, we also assayed growth at two different pH values (7.0 and 8.0) and in two different NaCl concentrations (170mM and 300mM) (Figs 1 and 2).

Growth of the wild type was similar across all of the pH and [Na$^+$] concentrations tested. Growth curves are shown in S1 Fig. Doubling times, calculated from early exponential growth, are in S1 and S2 Tables. There are clear differences in growth between LB and SCFM. In the minimal medium the doubling times generally are longer and the final cell density lower in SCFM. In comparison, changes in pH and [Na$^+$] had little systematic effect. However, at pH 8.0 and 300 mM NaCl, the doubling times in SCFM and LB are not statistically significantly different, and in these conditions, in both LB and SCFM, the growth curves show more than one phase.

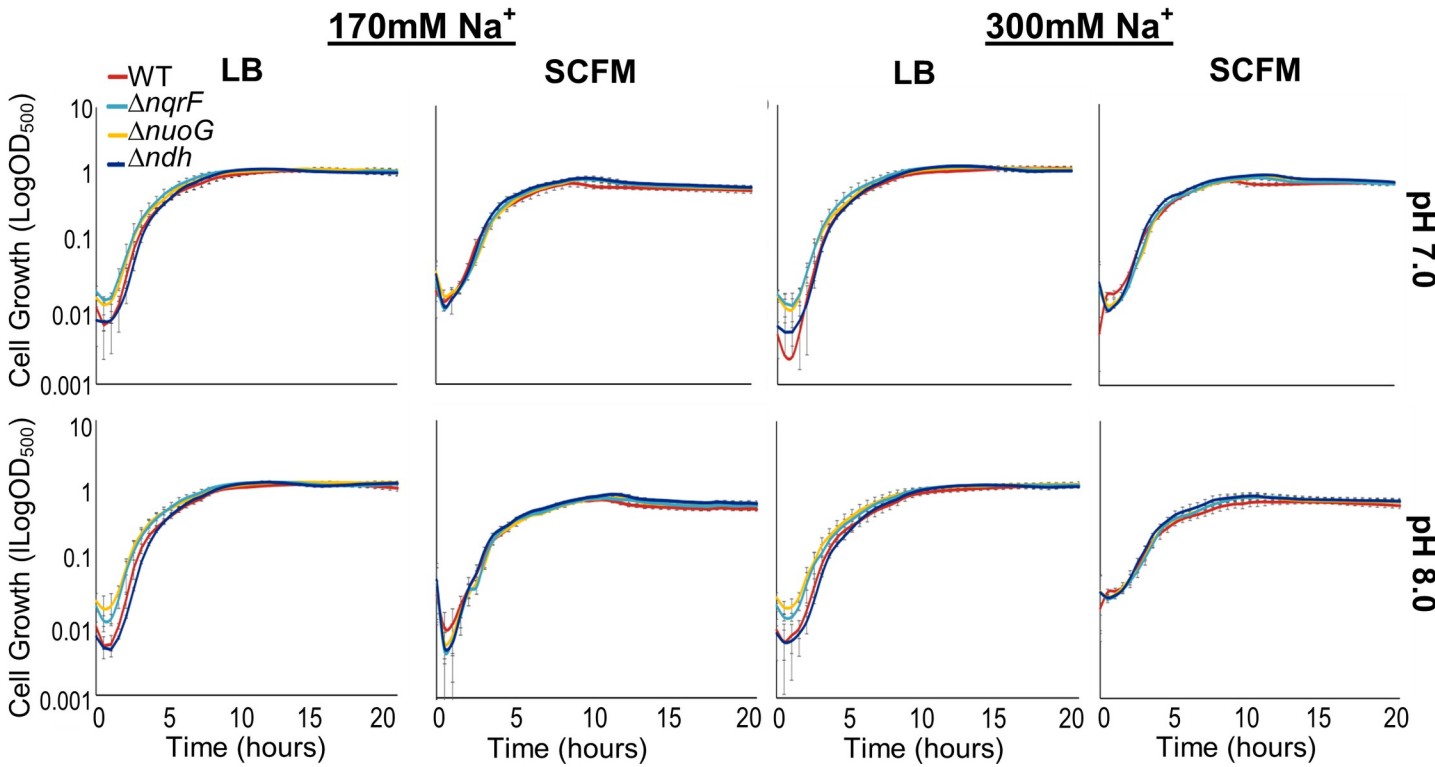

**Fig 1. Comparison of growth of wild type and single deletion mutant strains.** Growth curves of wild type PAO1 (red) and single deletion mutants Δ*nqrF* (cyan), Δ*nuoG* (yellow), and Δ*ndh* (dark blue) plotted in $\log_{10}$ scale in LB and SCFM media at 170mM and 300mM NaCl concentrations, at pH 7.0 and 8.0. Changes in $OD_{500}$ were measured using a Tecan Infinite M1000 Pro plate reader during 20 hours of growth at 37˚C with continuous orbital shaking at 217 rpm. Each curve was constructed using two biological replicates with three technical replicates each, with standard deviation calculated accordingly and represented as error bars.

Growth curves for the single-deletion mutants (Δ*nqrF*, Δ*nuoG*, Δ*ndh*) together with wild type are shown in Fig 1 and doubling times are shown in S1 and S2 Tables. All of the single-deletion mutants were able to grow well in both LB and SCFM and in all pH and [Na⁺] conditions. The doubling times were generally longer than for the wild type, but never twice as long.

For the double-deletion mutants (Δ*nuoG*Δ*ndh*), (Δ*nqrF*Δ*ndh)*, and (Δ*nqrF*Δ*nuoG)*, growth curves are shown in Fig 2, and doubling times calculated from early exponential growth are compiled in S1 and S2 Tables. As in the case of the single-deletion mutants, all of the double-deletion mutants were able to grow well. This indicates that after deletion of one or two of the three NADH dehydrogenases, the energetic pathways available to the cells are still capable of supporting robust growth. For both single- and double-deletion mutants, rates and extents of growth were sufficient that cell membranes could be obtained for subsequent biochemical analysis (see below). The responses of the wild type to changes in type of medium (rich vs. minimal), pH and [Na⁺] are still generally observed in the mutants. There are many small differences between results for wild type and the mutants. We note the following that could warrant future research: In all conditions of medium, pH and [Na⁺] the mutant with only NUO (Δ*nqrF*Δ*ndh*), underwent an extended lag phase after which it was able to grow strongly. In SCFM this mutant appears to grow in more than one phase, similar to what is observed in the wild type at pH 8 and 300 mM NaCl (above). In the mutant, this pattern is most marked at pH 8 where two distinct phases can be discerned, with a clear lag phase between (S3 Fig). The doubling time of the mutant with only NQR (Δ*nuoG*Δ*ndh*) shows a strong pH dependence in

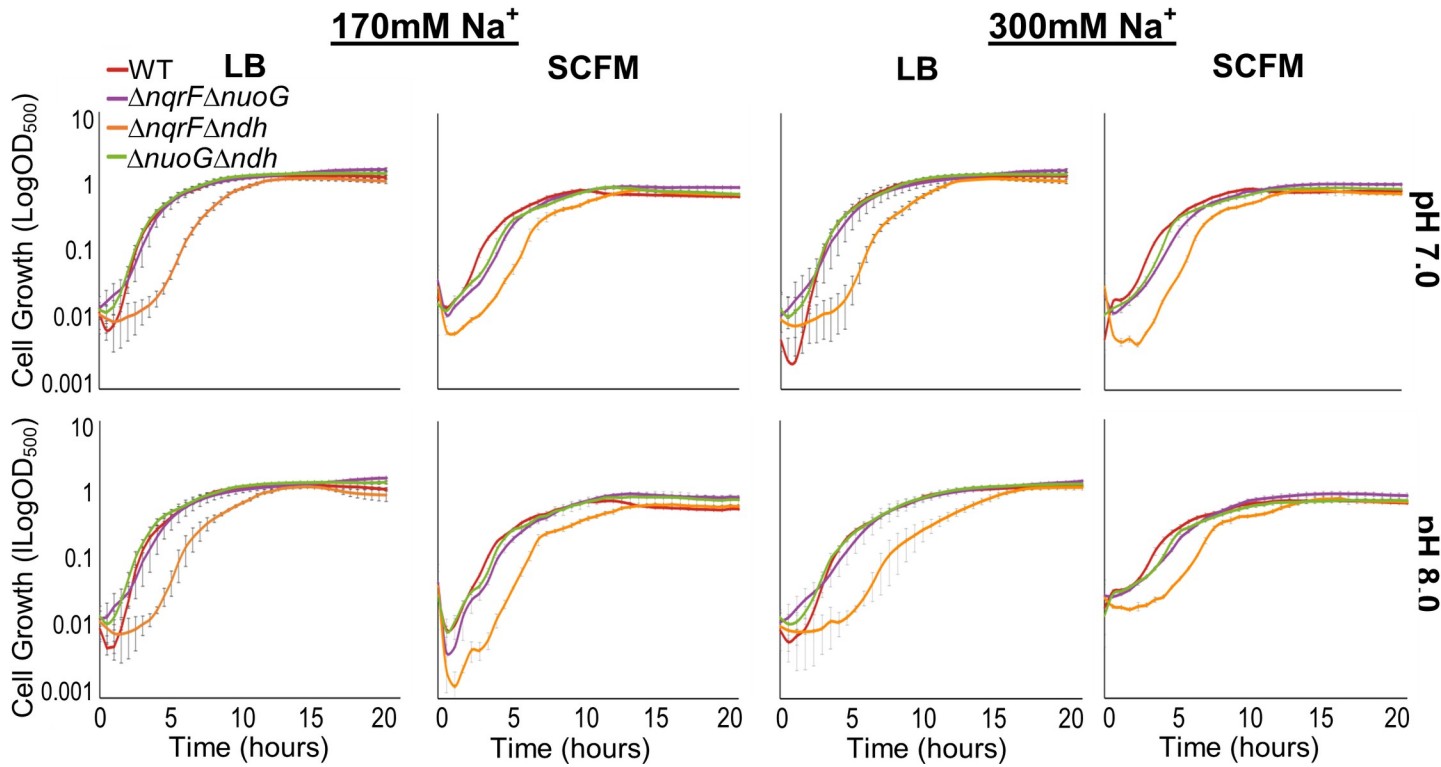

**Fig 2. Comparison of growth of wild type and double deletion mutant strains.** Growth curves of wild type PAO1 (red) and double deletion mutants Δ*nuoG*Δ*ndh* (green), Δ*nqrF*Δ*ndh* (orange), and Δ*nqrF*Δ*nuoG* (purple) plotted in log$_{10}$ scale in LB and SCFM media at 170mM and 300mM NaCl concentrations, at pH 7.0 and 8.0. Changes in OD$_{500}$ were measured using a Tecan Infinite M1000 Pro plate reader during 20 hours of growth at 37°C with continuous orbital shaking at 217 rpm. Each curve was constructed using two biological replicates with three technical replicates each, with standard deviation calculated accordingly and represented as error bars.

SCFM, but not in rich medium. The mutant with only NDH2 (Δ*nqrF*Δ*nuoG)* shows a dependence on [Na$^+$] in rich medium; if there is a similar trend in SCFM it is far weaker.

## NADH:quinone oxidoreductase activities in the double deletion mutants

Growth measurements show that any one of the three NADH dehydrogenases is capable of supporting growth. The activity measurements confirm that all of the double-deletion strains have significant NADH dehydrogenase activity in both exponential and stationary phases. For example, in the extreme case the mutant with only NUO has only 15% of wild type activity. To quantify the NADH:quinone oxidoreductase activity for each of the three NADH dehydrogenases we measured the enzyme activity in membranes from the double deletion strains, each of which has only one NADH dehydrogenase. For each mutant strain, membranes were prepared from cells harvested in both exponential and stationary phases. NADH:quinone oxidoreductase activity was measured by following the oxidation of NADH spectrophotometrically and the activities were normalized according to total membrane protein concentration. Saturating concentrations of NADH and ubiquinone-1 were used in order to obtain initial turnover rates close to Vmax (see Materials and Methods). Use of double-deletion strains is the clearest available approach to understanding the contributions of each of the NADH dehydrogenases. The only selective alternative substrate, deamino-NADH, was used to analyze activity in membranes from the single-deletion strains (below).

Fig 3A shows initial activities for exponential and stationary phase membranes. In all cases the membranes retained significant NADH dehydrogenase activity indicating that the one

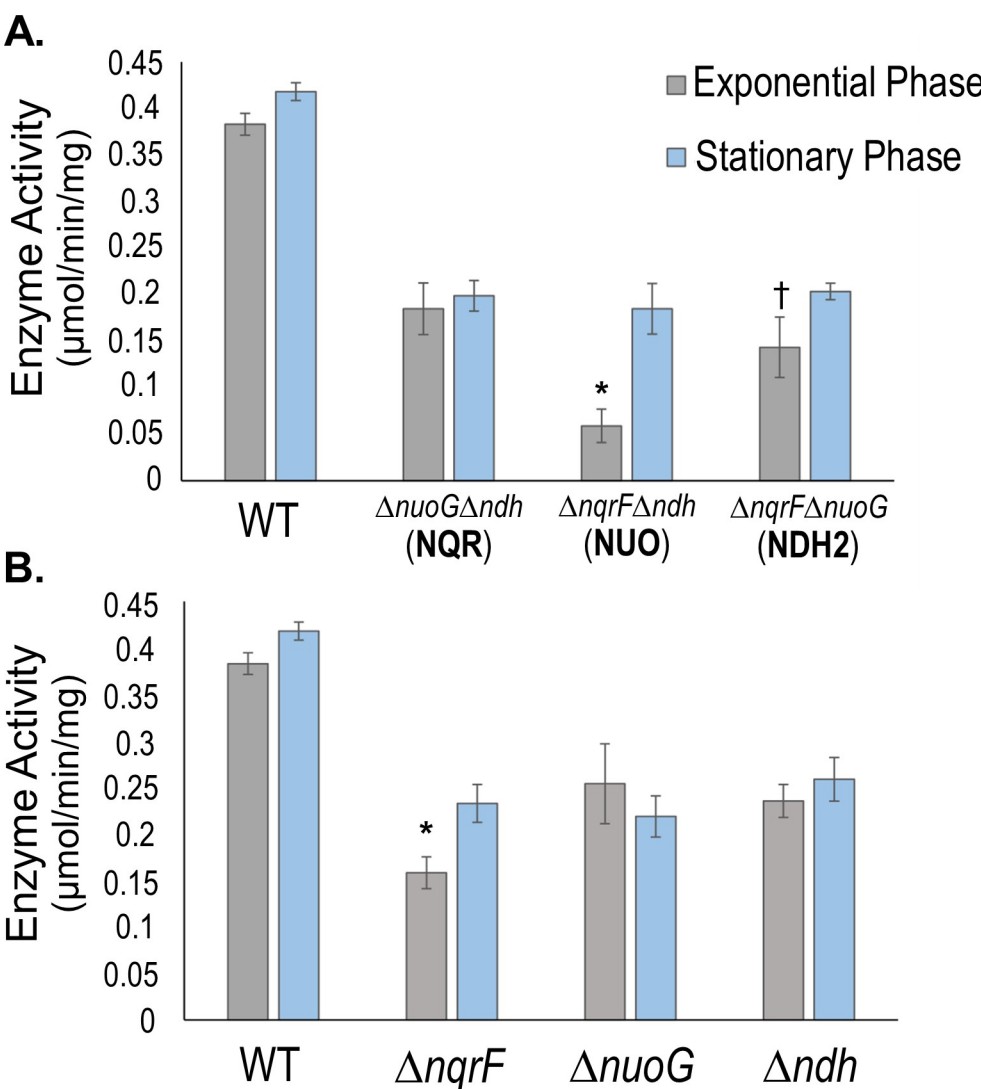

**Fig 3. NADH:quinone oxidoreductase activities of membranes from single and double deletion mutants compared to wild type. A**. Activity in membranes from wild type (WT) and double deletion mutants Δ*nuoG*Δ*ndh* (NQR only), Δ*nqrF*Δ*ndh* (NUO only), and Δ*nqrF*Δ*nuoG* (NDH-2 only) from cells harvested in exponential phase (grey) and stationary phase (blue). **B**. Activity in membranes from WT and single deletion mutants Δ*nqrF*, Δ*nuoG*, Δ*ndh* from cells harvested in the exponential phase (grey) and stationary phase (blue). Enzyme activity is defined as the μmoles of NADH consumed per minute per mg of membrane protein. The reaction contained: 100 μM NADH, 50 μM ubiquinone-1 and 100 mM NaCl. Changes in absorbance were followed at 340nm ($\varepsilon_{NADH}$ = 6.22 mM$^{-1}$ cm$^{-1}$). Activities for all mutant strains were significantly lower ($p \leq 0.01$) than WT in all conditions tested, according to a one-way ANOVA analysis. The Δ*nqrF*Δ*ndh* double mutant activity was significantly lower ($p \leq 0.01$, *) than the activities of the other two double mutants. The Δ*nqrF*Δ*nuoG* double mutant activity was significantly lower ($p \leq 0.05$, †) than the Δ*nuoG*Δ*ndh* mutant. The Δ*nqrF* single mutant activity was significantly lower ($p \leq 0.01$, *) than the activities of the other two single mutants.

remaining enzyme was expressed and active. Double deletion mutants showed significantly lower rates of NADH consumption than the wild type. In exponential phase, the mutant with only NUO had 15% of wild-type activity, while the mutants with only NDH2 and only NQR had 37% and 48%, respectively (S3 Table). In the case of stationary phase, the mutants with NQR, NUO, or NDH2 only had 48%, 46%, and 49% of wild-type activity, respectively (S3 Table). NADH dehydrogenase activity was consistently greater in membranes from stationary phase

cells than in exponential phase membranes, but these data suggest that each enzyme is actively contributing to the total NADH dehydrogenase activity in wild type in both growth phases.

## NADH:quinone oxidoreductase activities in single deletion mutants

We then carried out NADH dehydrogenase activity measurements on the three single deletion mutants, each of which retains two of the three enzymes. The results are summarized in Fig 3B. For all of the mutants, activity was lower than in the wild type, for membranes from both exponential and stationary phases. In the mutant that lacks an active NQR (Δ*nqrF*) the NADH dehydrogenase activity was 59% lower in exponential phase and 46% lower in stationary phase, compared to wild type (S3 Table). Of the three single deletion mutants, Δ*nqrF* had the greatest loss of activity in exponential phase, indicating that NQR is the major contributor to NADH dehydrogenase activity during exponential growth.

In the mutant lacking NDH2, activity in both exponential phase and stationary phase was approximately 62% relative to wild type, a decrease of 38% (S3 Table). This is consistent with the results from the double deletion mutant expressing only NDH2 (Δ*nqrF*Δ*nuoG*) which retained approximately 37% of NADH activity during exponential phase (Fig 3A). The contribution of NDH2 can also be assessed directly in membranes from wild-type cells by using deamino-NADH (dNADH) instead of NADH as a substrate. dNADH can be oxidized by NUO and NQR but not by NDH2 [45–47]. Membranes from wild type cells harvested in stationary phase had 38% lower activity with dNADH compared to NADH (S4 Fig). In membranes from wild type cells harvested in exponential phase, dNADH activity was 30% lower than NADH dehydrogenase activity. Although the almost exact correspondence of the numerical values for exponential phase membranes is probably fortuitous, these results are consistent in showing that NDH2 is responsible for a significant fraction of the total NADH dehydrogenase activity in both exponential and stationary phases.

## Consequences of deletion of NADH dehydrogenases for the physiology of *P. aeruginosa*: Pyocyanin production and biofilm formation

Two notable virulence traits of *P. aeruginosa* are (i) the production of biofilm and (ii) the secretion of pyocyanin. Both traits are controlled by quorum sensing and typically manifest themselves during stationary phase [41]. We noticed that some of the strains constructed for this study began to form biofilm earlier, and produced larger amounts of pyocyanin, than wild type. Based on these observations, we carried out a systematic study of biofilm formation and pyocyanin in these mutants.

**Biofilm formation.**   Production of biofilm was measured by tracking three different parameters: (i) crystal violet retention (a measure of total biofilm produced), (ii) biofilm thickness, and (iii) biofilm surface area (see Materials and Methods). Samples were analyzed at mid-attachment phase (6 hours) and maturity (24 hours) [40,48]. At mid-attachment, the crystal violet retention values for Δ*nqrF* and Δ*nuoG* were 1.55 AU and 1.43 AU, respectively, compared to 1.12 AU for the wild type (Fig 4). For both mutants, the difference from the wild type is statistically significant. However, at the 24-hour time point, there was no significant difference between the wild-type and mutant values. In contrast, in the Δ*ndh* mutant, the crystal violet assay showed less biofilm than the wild type at both 6-hour (0.94 AU) and 24-hour (1.68 AU) time points.

To assess biofilm thickness and area, two-photon fluorescence microscopy images of propidium iodide-stained biofilm were collected at 6- and 24-hour time points and analyzed using COMSTAT2 software [42,43]. At 6 hours, the biofilm formed by the Δ*nuoG* strain, was significantly thicker and covered a greater area than in the wild type. In the Δ*nqrF* and Δ*ndh* strains, biofilm thickness and surface area were both similar to wild type (Fig 4B, 4C and S3 Fig). At 24

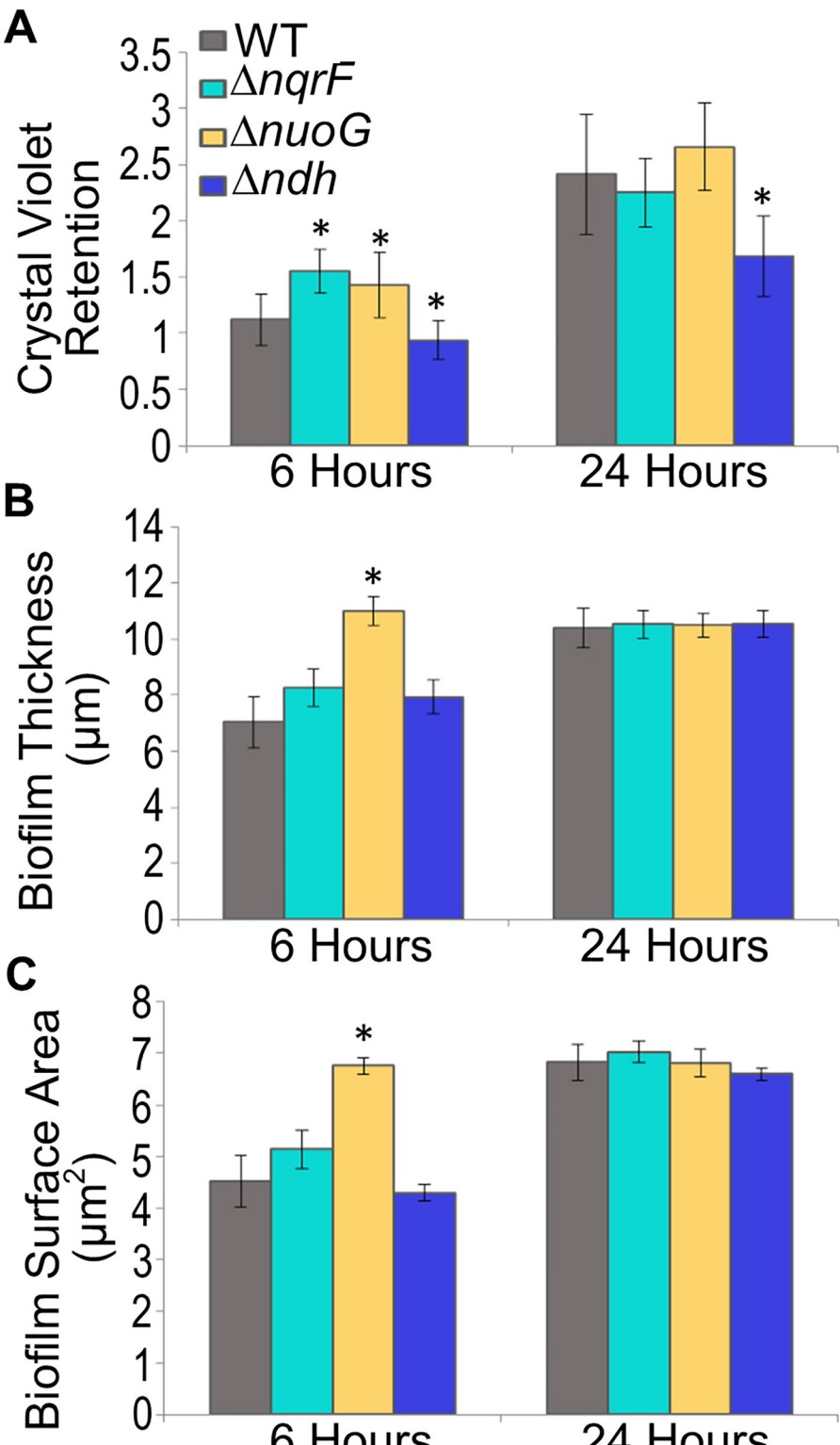

**Fig 4. Biofilm formation by wild type (PAO1) and single deletion mutants.** A. Crystal violet retention was measured as the total amount of biofilm produced. WT (grey), Δ*nqrF* (cyan), Δ*nuoG* (yellow) and Δ*ndh2* (dark blue) were diluted to $OD_{600} = 0.5$ from an overnight culture, and grown in a 96-well plate, without shaking for 6 h (mid-attachment) or 24 hours (mature biofilm). Crystal violet retention of the biofilm was examined at each time point through optical density at 600 nm. B. Surface area of the biomass from COMSTAT2 analysis of images taken with two-

photon microscope. For all panels, error bars indicate standard error of the mean from three biological replicates, and stars indicate p-values ≤ 0.05 compared to wild type through a student's *t*-test. (C) Average thickness of the biofilm was measured using propidium iodide (30 μM) stained biofilms grown on 8-well coverslips for 6 or 24 hours. The fluorescence of the retained propidium iodide in the biofilm was measured with a two-photon microscope. COMSTAT2 analysis was used for image analysis.

hours, none of the mutants was different from the wild type in biofilm thickness or surface area. In the Δ*ndh* strain, at 6- and 24-hours, biofilm thickness and surface area were very similar to wild type, but at both time points, the crystal violet assay showed significantly less retention. This suggests that the biofilms formed by this mutant, although extensive and thick, may be less dense.

**Pyocyanin secretion.** Pyocyanin is a toxin produced by *P. aeruginosa* that can readily diffuse to target cells, where it will cause oxidative stress. Pyocyanin is a vivid blue color and can be identified by its visible spectrum and measured on the basis of absorbance at 690 nm [49]. Pyocyanin production is controlled by quorum sensing and is thus associated with stationary phase. In wild-type *P. aeruginosa*, in rich medium, production of pyocyanin could be observed after 16 hours of growth and the concentration reached a maximum of 3.6 μM at 18 hours (Fig 5). In each of the single deletion mutants, pyocyanin production began earlier and reached much higher concentrations (Fig 5). The Δ*nqrF* strain produced the highest levels of pyocyanin (43.5 μM at 20 hours post inoculation), 12 times more than wild type.

## RNAseq transcriptome analysis of the strain lacking NQR (Δ*nqrF*)

To understand the physiological changes taking place in the mutant lacking NQR (Δ*nqrF*), we used RNA sequencing to compare their transcriptomes in both mid-exponential and stationary phases. Comparison of the mapped reads in Δ*nqrF* compared with wild type, revealed

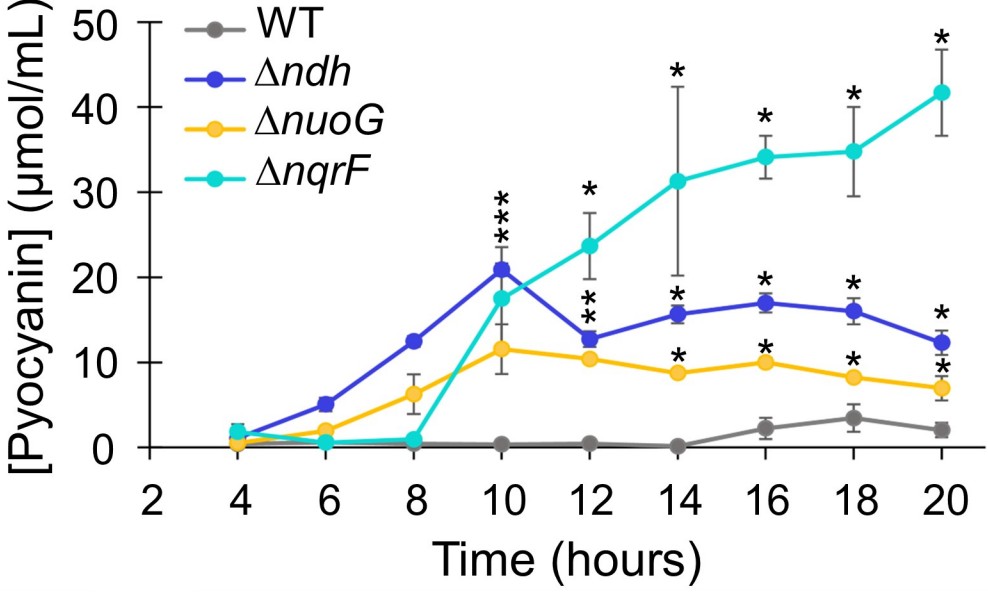

**Fig 5. Pyocyanin production.** Pyocyanin production in wild type PAO1 (grey), Δ*ndh* (dark blue), Δ*nuoG* (yellow) and Δ*nqrF* (cyan). The concentration (μM) was determined using the extinction coefficient for pyocyanin at 690 nm (4130 $M^{-1}cm^{-1}$). $OD_{600}$ of the wild type and mutant cultures were similar at each time point, so data represent an accurate comparison of pyocyanin production in the different strains. Stars indicate p-values from a student's *t*-test ≤ 0.05, compared to wild type at each time point.

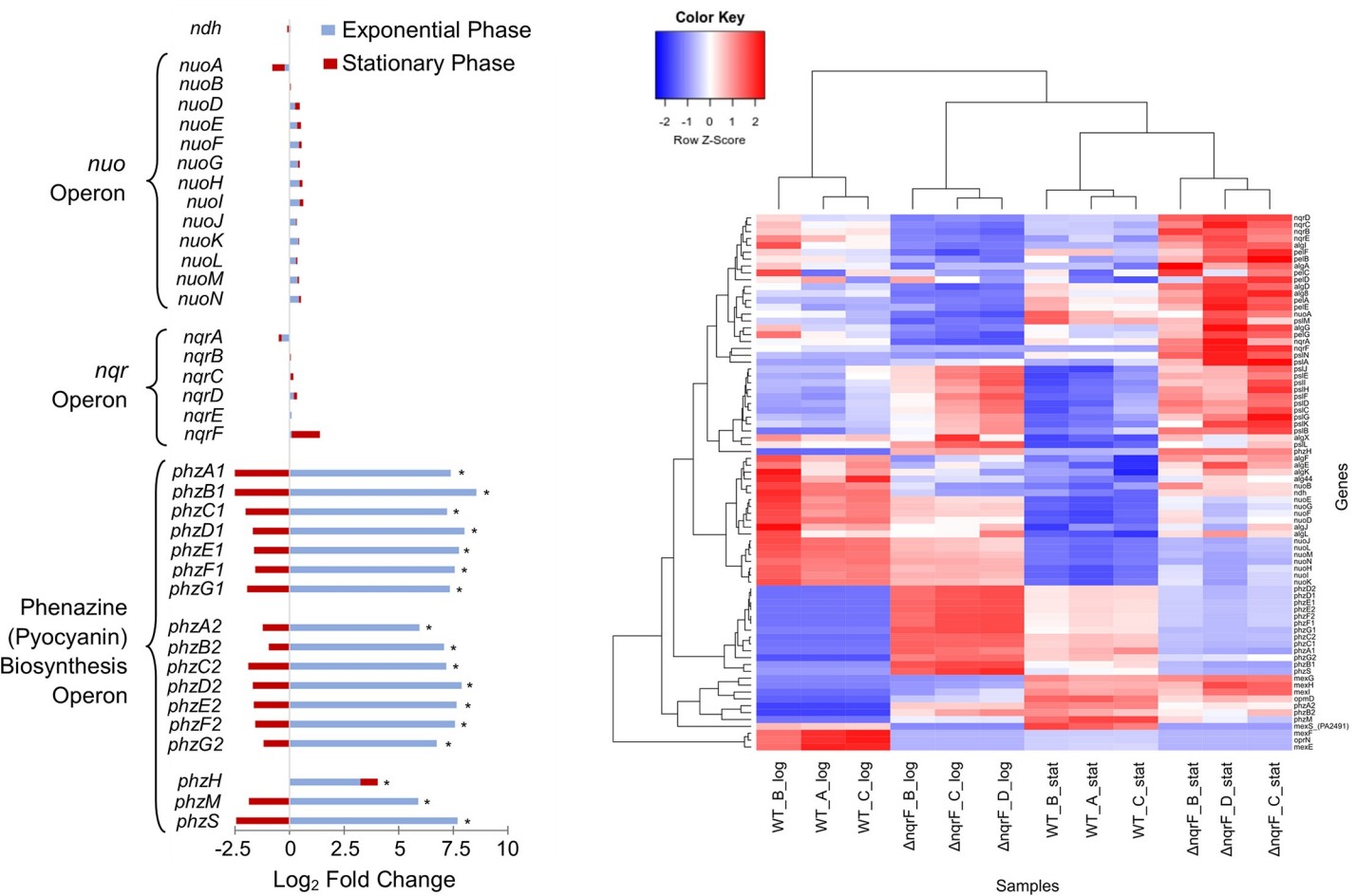

**Fig 6. Transcriptome analysis of the ΔnqrF mutant strain compared to wild type.** A. Differential expression of genes encoding NDH2, NUO, and the pyocyanin biosynthesis operons in the ΔnqrF mutant strain compared with wild type PAO1 during exponential and stationary phases. A positive value indicates an increased expression in ΔnqrF compared to wild-type. Stars indicate significant changes in expression with a P-value <0.05 as determined via the calculations outlined in Materials and Methods. B. Heatmap of gene expression profiles in the transcriptome profiles of three biological replicates of the ΔnqrF and wild type strains in exponential and stationary phase. Select genes include those encoding *ndh*, *nuo operon*, pyocyanin biosynthesis ron, extracellular polysaccharides *alg*, *psl*, and *pel* operons, the genes coding for the MexEF-OrpN efflux pump and related genes and the MexGHI-OpmD efflux pump. All genes were normalized by gene length and $\chi^2$ transformed before analysis. Most highly expressed genes were selected according to the sum of normalized counts across samples. Genes were selected from the NOISeq results and counts were $\chi^2$ transformed before analysis. According to the Z-score, color bars in red correspond to higher expression.

significant changes in the transcription of the genes coding for enzymes that make up the pyocyanin synthesis pathway. In exponential phase, the expression of each of the pyocyanin synthesis genes were ~6–8.5-fold greater in ΔnqrF (Fig 6A and 6B), a result which is consistent with the observed increase in pyocyanin production by this strain. Expression of the remaining NADH dehydrogenases NUO and NDH2 was shown not to change in ΔnqrF relative to wild type at either time point, indicating that the absence of a functional NQR does not affect the transcription of the other two enzymes.

## Changes in antibiotic resistance in single mutants

*P. aeruginosa* is a multi-drug resistant pathogen with a wide range of intrinsic antibiotic resistance. Previous studies have shown that disruptions of NUO and NQR in *P. aeruginosa* result in increased resistance to the aminoglycosides gentamicin and tobramycin [22,50]. We examined the resistance of the three NADH dehydrogenase single deletion mutants to the

antibiotics available in our laboratory, and our results reveal that Δ*nqrF*, Δ*nuoG* and Δ*ndh* all have different resistance profiles. Table 3 shows the minimum inhibitory concentrations (MIC) for the antibiotics tested in each of the single mutant strains and wild type. Relative to wild type, all the mutants showed higher sensitivity to chloramphenicol and trimethoprim, but lower sensitivity to kanamycin and gentamicin, each to a different degree. In the case of streptomycin, Δ*nqrF* and Δ*ndh* show lower sensitivity, while Δ*nuoG* is unchanged.

## Infection in cellular and animal models

The enhanced pyocyanin production we observed, suggested that these strains might also have altered virulence. We therefore tested virulence in two well established model systems: macrophages and mice. In macrophages, compared to a bacteria-free control, infection by the wild type, or any of the three mutant strains, produced at least 3-times as much cell death (Fig 7). The cellular death rate caused by Δ*nuoG* was approximately the same as wild-type, while Δ*nqrF* and Δ*ndh* were able to kill macrophages 5- and 7-times more efficiently respectively.

Given that the Δ*nqrF* strain exhibits the greatest killing capacity in the macrophage assay, we assessed whether this mutant was also defective in virulence in mice. The Δ*nqrF* strain was compared to wild type PAO1 in an intranasal aspiration mouse model of acute pneumonia. Mice infected with ~4 x $10^6$ CFU of the Δ*nqrF* mutant progressed to prelethal illness more rapidly than those infected with wild-type PAO1 ($p < 0.05$) (Fig 8A, S7 Fig). A second experiment was performed to assess the bacterial load in the lungs of infected mice. To avoid rapid death of the mice, a lower bacterial inoculum (~5–6 x $10^5$ CFU) was given to the mice, which were then euthanized 24 hours after infection. The lungs were sterilely removed, homogenized, and plated to measure bacterial loads. We found that the bacterial load was approximately the same in mice infected with wild type PAO1 and Δ*nqrF* (Fig 8B). Thus, the higher lethality of Δ*nqrF* is due to inherent virulence and not simply a better ability to colonize and proliferate in the lungs.

## In trans complementation of NQR

To confirm that the observed phenotype of the Δ*nqrF* strain is a direct effect of NQR inactivation, we created a complementation strain (pHERD28C-NQR), using the complete *nqr* operon, with a C-terminal histidine tag, in the pHERD plasmid, under control of the arabinose (*ara*) promotor. Membranes isolated from this strain showed essentially the same NADH oxidation activity as wild type (Fig 9A). After partial purification on a Ni-NTA column, the preparation was run on an SDS gel, where it showed the two fluorescent bands characteristic of the FMN cofactors of the NqrB and NqrC subunits (Fig 9B–9D) [29]. The strain also showed a

**Table 3. Minimal Inhibitory Concentrations (MIC)[a] for wild type (PAO1), Δ*nqrF*, Δ*nuoG*, and Δ*ndh*.**

| Antibiotic | WT | Δ*nqrF* | Δ*nuoG* | Δ*ndh* |
|---|---|---|---|---|
| MIC (mg/mL) | | | | |
| Streptomycin | 35 | 90 | 35 | 240 |
| Chloramphenicol | 300 | 60 | <50 | 35 |
| Kanamycin | 90 | 175 | 140 | >1000 |
| Trimethoprim | 900 | 160 | 90 | 300 |
| Gentamicin | 3 | 175 | 5 | >1000 |

[a]MICs were determined by inoculating 4 μL of an overnight culture into 1 mL LB + appropriate antibiotic in a 48-well plate. Plates were incubated at 37°C, in aerobic conditions for 24 hours before determining MIC.

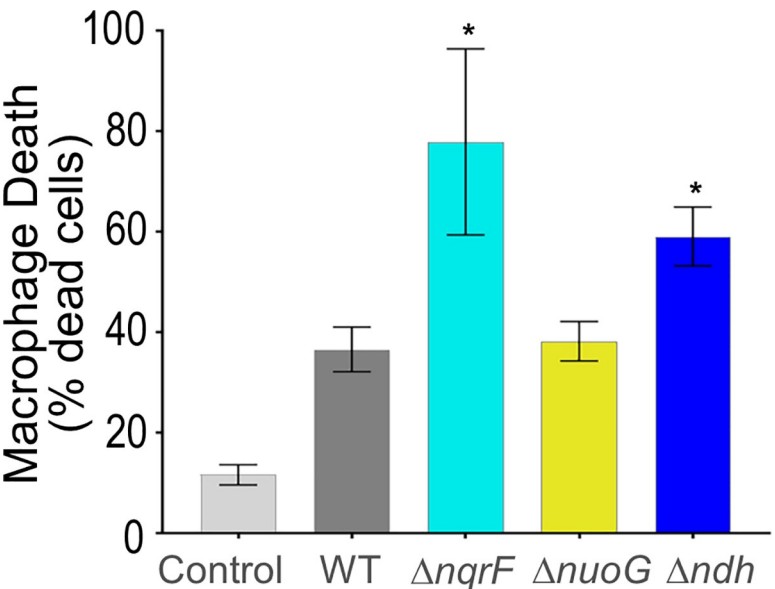

**Fig 7. Macrophage toxicity assay.** Wild type PAO1, Δ*nqrF*, Δ*nuoF* and Δ*ndh* mutants were cocultured in a 1:1 ratio with RAW 264.7 macrophages for 6 hours before a LDH assay was performed to determine macrophage cell death. Data from three separate experiments are represented as the average % of cell death compared to macrophage death of the bacteria-free control from that experiment, with error bars indicating the standard error of the mean. Stars indicate p-values $\leq 0.05$ compared to wild type, through a student's t-test.

reversion towards wild-type characteristics of biofilm formation and pyocyanin production. As described above, the deletion strain lacking NQR, shows enhancement of some properties associated with virulence: biofilm formation begins earlier than in the wild type and the cells produce much more pyocyanin. Crystal violet retention analysis of mid-attachment phase

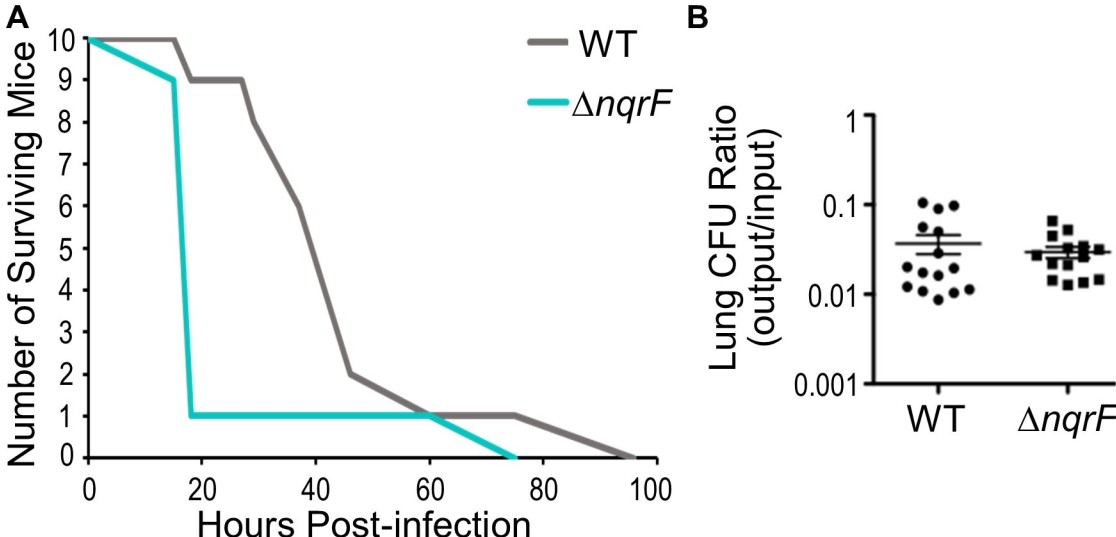

**Fig 8. Survival of mice infected with wild type PAO1 and Δ*nqr*.** A. Survival of BALB/c mice after intranasal aspiration of ~4 x10$^6$ CFU wild type or Δ*nqrF*. The survival curves were statistically different ($p < 0.05$; log rank test). B. Bacterial burden in the lungs. Lungs were harvested from mice infected with ~5–6 x 10$^5$ CFU at 24 h post-infection, and the ratio of CFUs before inoculation and after death were counted. There is no significant difference between the bacterial burden from wild type PAO1 and Δ*nqrF*. (This is representative of two independent experiments).

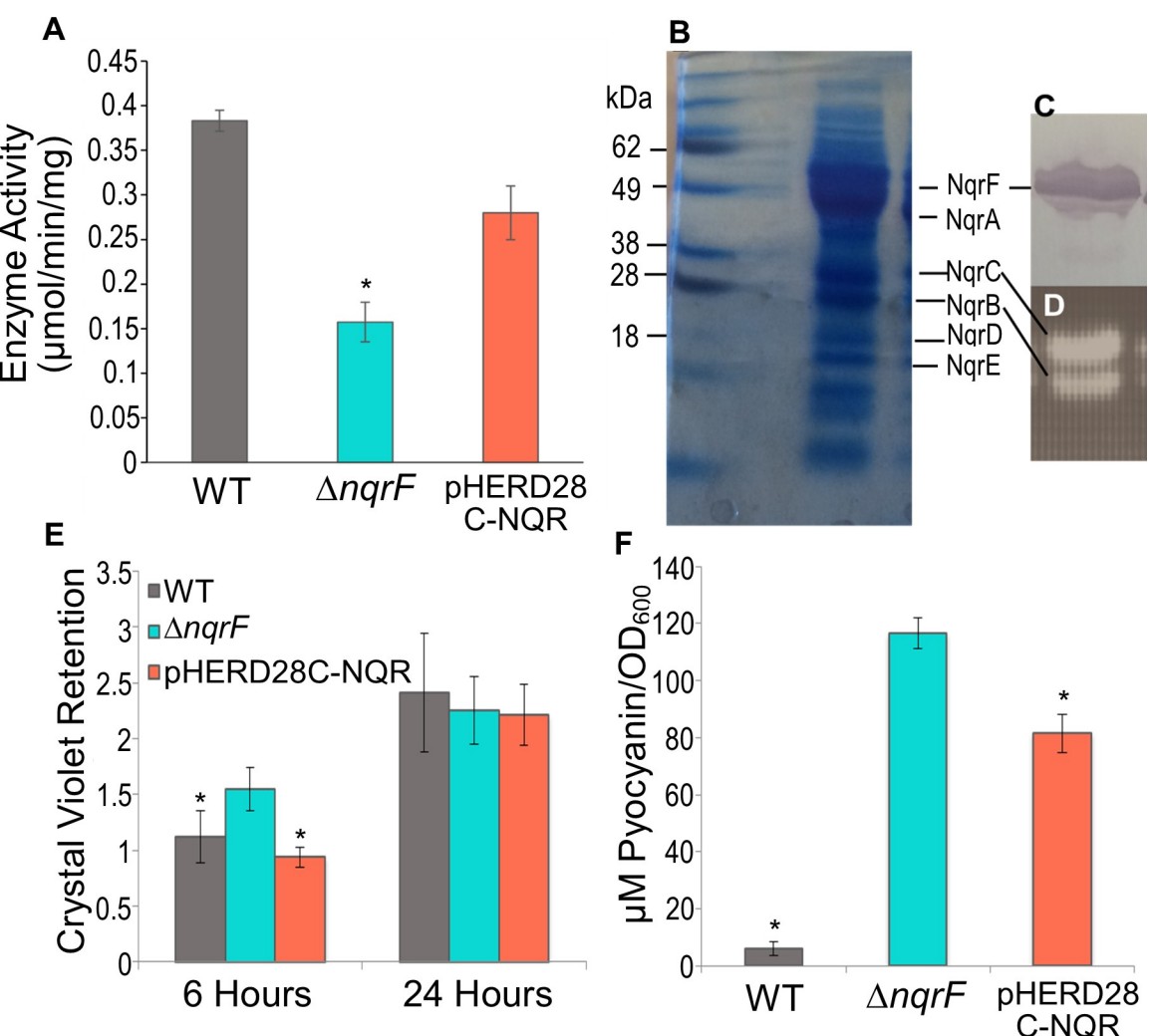

**Fig 9. Complementation of NQR.** A. NADH:quinone oxidoreductase activities of wild type (PAO1), Δ*nqrF*, and *nqr*-complementation strain (pHERD28C-NQR). Enzyme activity is defined as the μmoles of NADH consumed per minute per mg of membrane protein. The reaction contains: 100 μM NADH, 50 μM ubiquinone-1 and 100 mM NaCl. Changes in absorbance were followed at 340nm ($\varepsilon_{\text{NADH}}$ = 6.22 $\text{mM}^{-1}$ $\text{cm}^{-1}$). and NADH dehydrogenase deletion mutants. B. SDS-gel (4–12%) of partially purified NQR complex expressed in the pHERD 28C-NQR (20 mg) run in Tris-glycine gel system stained with Coomassie Blue. C. Western blotting using anti-His5X antibodies showing the NqrF subunit, where the histidine tag is attached. D. Same gel as in (B) exposed to UV light before staining showing the fluorescent bands corresponding to the NqrB and NqrC subunits of the semi-purified NQR. E. Biofilm formation quantified as crystal violet retention at mid-attachment (6 hours) and mature (24 hours) biofilm. F. Pyocyanin concentration (μM) following 24 hours of growth, measured at 690 nm (4130 $\text{M}^{-1}\text{cm}^{-1}$) and normalized to culture $OD_{600}$. $OD_{600}$ of the wild type and mutant cultures were similar at each time point, ensuring that the data represent an accurate comparison of pyocyanin production between strains. Stars indicate p-values from a student's *t*-test $\leq$ 0.05.

samples revealed a significant decrease in measured biofilm production by the pHERD28C-NQR strain relative to that of Δ*nqrF*, representing a complete return to wild type characteristics (Fig 9E).

In the case of pyocyanin production, at 24 hours post inoculation, pHERD28C-NQR produced an average of 85 μM pyocyanin/$OD_{600}$ (Fig 9F). This is a significant decrease from the 117 μM of pyocyanin/$OD_{600}$ produced by Δ*nqrF* at the same time point, but still much more than produced by wild type. Even so, the reversion toward wild type characteristics upon reintroduction of NQR confirms that it is the deletion of NQR which leads to the mutant phenotype.

## Discussion

In this study we have characterized the roles of the three NADH:quinone oxidoreductases of *Pseudomonas aeruginosa*: NQR, NUO, and NDH2. We used single- and double-deletion mutants to assess the importance of each enzyme in the growth of *P. aeruginosa* and the contribution of each to the net NADH dehydrogenase activity. Our results suggest connections between the deletion of these enzymes and changes usually associated with the transition from exponential to stationary phase, including biofilm formation and pyocyanin production. Since these traits are known to be associated with virulence, this led to the discovery that some of the deletion mutants have enhanced lethality in infection models with both isolated macrophages and mice.

### Effect of single and double NADH dehydrogenase deletions on growth

None of the single NADH dehydrogenase deletion mutants caused an insurmountable defect in growth in any of the conditions we tested, including rich medium (LB) and minimal medium with glucose (SCFM), and at low and high pH and low and high [Na$^+$].

The double deletion mutants were constructed with the aim of having strains in which only one of the three NADH dehydrogenases would be present so that the contribution of each enzyme could be evaluated separately. All three double-deletion mutants were able to grow in all conditions tested. The only notable difference was in the mutant that has only NUO (Δ*nqrF*Δ*ndh*), which had an extended lag phase lasting several hours, after which growth proceeded normally. Torres et al. reported a similar lag in the growth of their double deletion mutant which expressed NQR only (Δ*nuoIJ*Δ*ndh*) [22]. In our hands the corresponding mutant (Δ*nuoG*Δ*ndh*) did not show this lag and grew normally. We do not have an explanation for this difference but taken together the results point to the conclusion that *P. aeruginosa* is able to grow fairly normally even when it has only one of the three NADH dehydrogenases. This reveals a remarkable robustness in the energy production systems of this bacterium.

### Assessing the contributions of NQR, NUO and NDH2 in exponential phase: NADH dehydrogenase activity in isolated membranes

In contrast to the growth curve analysis, where none of the mutants caused a pronounced growth defect, when NADH consumption was measured in isolated membranes, all of the mutants, single- and double-deletion, showed lower levels of activity (normalized to total protein) compared to the wild type. In the single mutants that lack: NQR (Δ*nqrF*), NUO (Δ*nuoG*) and NDH2 (Δ*ndh*), initial rates of NADH oxidation were 41%, 67%, and 62%, respectively, compared to wild type (S3 Table). In the double deletion mutants that have only: NQR (Δ*nuoG*Δ*ndh*), NUO (Δ*nqrF*Δ*ndh*) and NDH2 (Δ*nqrF*Δ*nuoG*), initial rates were 48%, 15%, and 37% respectively (S3 Table). When activity was measured in the wild type, using deamino-NADH, which does not react with NDH2, the initial rate was 70% compared to the reaction of wild type with NADH (S4 Fig). In the reaction with deamino-NADH, only NQR and NUO should be active, so this is comparable to the activity of Δ*ndh* mutant with NADH: 62% compared to wild type. The activity of wild type with deamino-NADH is 8% greater than the activity of the mutant lacking NDH2 (Δ*ndh*). One possible explanation is that, although deamino-NADH is described as not reactive with NDH2, there may actually be a small amount of residual activity with this substrate [51].

One significant conclusion from these results is that, during exponential phase, out of the three NADH dehydrogenases, NQR appears to make the largest contribution to activity. Among the single-deletion mutants, the strain without NQR (Δ*nqrF*) shows the largest relative

decrease in activity (59% decrease) (S3 Table). Similarly, among the double-deletion mutants, the two strains that have lost NQR (Δ*nqrF*Δ*ndh* and Δ*nqrF*Δ*nuoG*) show the largest loss of activity (85% and 63% decrease respectively). Taken together, these results show that, during exponential growth in *P. aeruginosa*, NQR likely accounts for the majority of NADH dehydrogenase activity. This finding is in agreement with the conclusions of Liang et al. on the basis of in-gel activity measurements [23]. Torres et al. also studied a series of NADH dehydrogenase deletion mutants in *P. aeruginosa* but concluded that NQR has only a minor role [22]. This difference is likely due to the fact that their measurements were made in essentially sodium-free conditions, where we would expect the activity of NQR to be very low, since this enzyme is Na$^+$-dependent.

### Assessing the contributions of NQR, NUO and NDH2 in stationary phase: NADH dehydrogenase activity in isolated membranes

The same set of measurements was also carried out in membranes from cells harvested during stationary phase. Initial rates of NADH oxidation, for single deletion strains lacking: NQR (Δ*nqrF*), NUO (Δ*nuoG*) and NDH2 (Δ*ndh*), were 54%, 53%, and 62% respectively compared to wild type (S3 Table). Initial rates in the double deletion mutants, that have only NQR (Δ*nuoG*Δ*ndh*), NUO (Δ*nqrF*Δ*ndh*) and NDH2 (Δ*nqrF*Δ*nuoG*), were 49%, 46%, and 48% respectively (S3 Table). These results do not point to any one of the enzymes as having a predominant role in NADH dehydrogenase activity during stationary phase. In the wild type, when deamino-NADH was used, activity was 62% compared to the reaction with NADH, consistent with a 38% contribution by NDH2. This compares to a loss of 51% of NADH activity in the strain where only NDH2 is present. This discrepancy is likely due to the fact that deamino-NADH is a less efficient substrate [51]. This again suggests that the effects of the mutants on the physiology of the bacteria go beyond simply subtracting one or more enzymes.

### Consequences of deletion of NADH dehydrogenases for the physiology of *P. aeruginosa*: Pyocyanin production

When we first began to grow the NADH dehydrogenase mutants we noticed a striking blue color in the liquid cultures. In stationary phase, *P. aeruginosa* cultures typically take on a light blue hue, but the color we observed appeared earlier and was much more intense. The blue chromophore was easily identified as pyocyanin on the basis of its visible spectrum. Pyocyanin is a toxin, produced only by *P. aeruginosa*, that acts by causing oxidative stress in neighboring cells [52]. Its production is controlled by quorum sensing and usually starts in stationary phase, but all of the NADH dehydrogenase mutants began making pyocyanin earlier, during exponential growth phase, and in much larger concentrations than wild type cells. Pyocyanin production begins first in the strain without NDH2 (Δ*ndh*), about 6 hours after inoculation, but reaches the highest levels—about 12 times as much as wild type—in the strain without NQR (Δ*nqrF*). The final step in the pyocyanin synthesis pathway is catalyzed by the enzyme PhzS in a reaction that requires NADH as a reductant [53]. Our RNASeq analysis showed that, in the mutant without NQR (Δ*nqrF*), expression of genes coding for the pyocyanin synthesis pathway, including *phzS*, were elevated as much as 8.5-fold compared to wild type. The elevated pyocyanin production in the deletion mutants could be due to their diminished NADH dehydrogenase capacity or to the higher levels of PhzS, or both. Complementation of the strain lacking NQR (Δ*nqrF*) with the *nqr* operon on a plasmid, resulted in a decrease in pyocyanin production, but it did not fully restore the levels in the wild type. This suggests that gene dosage effects are playing a role, since the plasmid is likely present in multiple copies. Also,

expression is under the control of an arabinose promoter. Either of these could cause higher than normal levels of production.

## Changes in antibiotic resistance in single NADH dehydrogenases mutants

Each of the single deletion mutants showed increased sensitivity to chloramphenicol and tri-methoprim. This is likely due to changes in expression of an antibiotic efflux pump. Our RNA-Seq analysis of the Δ*nqrF* mutant showed that expression of MexEF-OprN, an efflux pump related to resistance against these antibiotics, is down-regulated 9-fold in exponential phase and 5-fold in stationary phase (S4 Table) [54,55]. The mechanism of this connection between deletion of the gene coding for NQR and changes in expression of MexEF-OprN is not known.

All of the single deletion mutants also showed increased resistance towards kanamycin and gentamicin, and Δ*nqrF* and Δ*ndh* showed increased resistance towards streptomycin. The RNASeq data on Δ*nqrF* do not suggest a clear mechanism for these changes. However, alterations in antibiotic resistance related to NADH metabolism in *P. aeruginosa* have been reported elsewhere. Aminoglycosides, such as kanamycin, gentamycin and streptomycin depend on the proton motive force to enter the cell [56–59]. Some of the NADH dehydrogenase deletion mutants may affect ion transport providing a possible explanation for our results.

## Infection models

Since pyocyanin is a virulence factor, we decided to evaluate whether the NADH dehydrogenase mutations caused changes in the ability of *P. aeruginosa* to infect cells and the outcomes of the infections. For this, we chose two well-characterized models of *P. aeruginosa* infection: macrophages in cell culture and whole mice.

In the case of macrophages, single deletion strains without NQR (Δ*nqrF*) and NDH2 (Δ*ndh*) were both significantly more effective in killing macrophages, while the strain without NUO (Δ*nuoG*) was approximately the same as wild type. Although the current work does not establish a direct causal connection between pyocyanin and the death of macrophages, it is worth noting that effectiveness in killing correlates with the total amount of pyocyanin produced rather than the time when pyocyanin production begins. Although pyocyanin production begins earlier in the strains that lack NUO (Δ*nuoG*) and NDH2 (Δ*ndh*), the strain that lacks NQR (Δ*nqrF*) produces the highest concentrations of pyocyanin and is also the most effective in killing.

For tests with mice, we chose to use the single deletion mutants that was most effective in killing macrophages: the strain that lacks NQR (Δ*nqrF*). Tests were carried out using an acute pneumonia infection model which tracked progression to prelethal illness over time. The strain lacking NQR (Δ*nqrF*) killed the mice at a significantly faster rate than wild type. However, at 24 hours after infection, the bacterial loads in the lungs of mice infected with either mutant or wild type were approximately the same. This suggests that the increased lethality of the strain lacking NQR (Δ*nqrF*) is not due to faster, or more efficient, proliferation in the mouse lungs, but rather to enhanced virulence of the individual cells. It is possible that this is due to the higher pyocyanin production in the strain. Pyocyanin has been shown to induce neutrophil apoptosis and play a critical role in the establishment and success of airway infections in mice [60,61]. Strains deficient in pyocyanin have been found to be cleared more rapidly from the lung and result in higher levels of recovery relative to wild type [60,61].

This putative role of pyocyanin does not appear to apply to all *P. aeruginosa* model-host systems. For example, inactivation of pyocyanin production had no effect on the virulence of *P.*

*aeruginosa* on the silkworm *Bombyx mori* [62]. Similarly, Torres et al. tested the virulence of their *P. aeruginosa* NADH dehydrogenase deletion mutants in two model systems: a plant (lettuce) and an insect (*Galleria mellonella*) and found no significant changes [22].

The overall picture that emerges from this study is consistent with the conclusions from a series of studies on unusually-virulent strains of *P. aeruginosa* found in CF patient samples. They found that many high virulent strains produced pyocyanin and other virulence factors (including LasA protease; see S4 Table) earlier and in larger quantities than the less virulent strains. However, the cause of this variation was attributed to changes in the quorum sensing control mechanisms [63–65]. The current work suggests that there may also be a connection to energy metabolism and NADH utilization. We plan to further test this hypothesis by constructing a series of double mutants in which the genes involved in pyocyanin synthesis have been deleted from strain that lack one or more NADH dehydrogenases. These double mutants will be tested for virulence, in comparison with wild type and a pyocyanin single deletion mutant. We expect that, if pyocyanin is the direct cause of enhanced virulence in the NADH dehydrogenase mutants, these double mutants would be no more virulent than wild-type *P. aeruginosa*.

## Supporting information

**S1 Fig.** Growth curves of wild type PAO1 in LB (A) and SCFM (B). Changes in $OD_{500}$ were measured using a Tecan Infinite M1000 Pro plate reader during 20 hours of growth at 37°C with continuous orbital shaking at 217 rpm. Each curve was constructed using two biological replicates with three technical replicates each, with standard deviation calculated accordingly and represented as error bars.
(TIF)

**S2 Fig. Comparison of growth of wild type and single deletion mutant strains.** Growth curves of wild type PAO1 (red) and single deletion mutants Δ*nqrF* (cyan), Δ*nuoG* (yellow), and Δ*ndh* (dark blue) in LB and SCFM media at 170mM and 300mM NaCl concentrations, at pH 7.0 and 8.0. Changes in $OD_{500}$ were measured using a Tecan Infinite M1000 Pro plate reader during 20 hours of growth at 37°C with continuous orbital shaking at 217 rpm. Each curve was constructed using two biological replicates with three technical replicates each, with standard deviation calculated accordingly and represented as error bars.
(TIF)

**S3 Fig. Comparison of growth of wild type and double deletion mutant strains.** Growth curves of wild type PAO1 (red) and double deletion mutants Δ*nuoG*Δ*ndh* (green), Δ*nqrF*Δ*ndh* (orange), and Δ*nqrF*Δ*nuoG* (purple) in LB and SCFM media at 170mM and 300mM NaCl concentrations, at pH 7.0 and 8.0. Changes in $OD_{500}$ were measured using a Tecan Infinite M1000 Pro plate reader during 20 hours of growth at 37°C with continuous orbital shaking at 217 rpm. Each curve was constructed using two biological replicates with three technical replicates each, with standard deviation calculated accordingly and represented as error bars.
(TIF)

**S4 Fig. NADH:quinone oxidoreductase activity in wild type (PAO1) using NADH and deamino-NADH as a substrates.** Membranes were harvested in exponential and stationary phase. Enzyme activity is defined as the μmoles of NADH (or deamino-NADH) consumed per minute per mg of membrane protein. The reaction contained: 100μM NADH, 50μM ubiquinone-1 and 100mM NaCl. Changes in absorbance were followed at 340nm ($\varepsilon_{NADH}$ = 6.22 mM$^{-1}$ cm$^{-1}$). Stars indicate p-values of $\leq 0.01$ compared to WT according to student's t-test.
(TIF)

**S5 Fig. Surface images of biofilm at mid-attachment.** Representative surface images of biofilm produced by WT (A), Δ*ndh* (B), Δ*nuoG* (C), and Δ*nqrF* (D) at the 6-hour timepoint.
(TIF)

**S6 Fig. Surface images of mature biofilm.** Representative surface images of biofilm produced by WT (A), Δ*ndh* (B), Δ*nuoG* (C), and Δ*nqrF* (D) at the 24-hour timepoint.
(TIF)

**S7 Fig. Survival of mice infected with wild type PAO1 and Δ*nqrF* in replicate experiment.** Survival of BALB/c mice after intranasal aspiration of ~4 x10^6 CFU wild type or Δ*nqrF*. The survival curves were statistically different ($p < 0.05$; log rank test).
(TIF)

**S1 Table. Doubling times[a] for WT PAO1, single, and double NADH dehydrogenase deletion mutants in LB medium[b].** [a] Maximum growth rates and doubling times were determined using the fit_easylinear algorithm provided in the growthrates R package [32, 33]. [b] Above parameters calculated from growth curves depicted in Figs 1 and 2. * indicates *P*-value ≤ 0.01, ** indicates *P*- value ≤ 0.001.
(DOCX)

**S2 Table. Doubling times[a] for WT PAO1, single, and double NADH dehydrogenase deletion mutants in SCFM medium[b].** [a] Maximum growth rates and doubling times were determined using the fit_easylinear algorithm provided in the growthrates R package [32, 33]. [b] Above parameters calculated from growth curves depicted in Figs 1 and 2. * indicates *P*-value ≤ 0.01, ** indicates *P*- value ≤ 0.001.
(DOCX)

**S3 Table. NADH Dehydrogenase Activity[a].** [a]Activities for each strain represented as a percentage of wild-type activity (100%)
(DOCX)

**S4 Table. Differential gene expression in the Δ*nqrF* mutant compared to WT in exponential and stationary phases[a].** [a]A positive value indicates an increased expression in Δ*nqrF* compared to wild-type. P-values were determined using the calculations outlined in Materials and Methods.
(DOCX)

**S1 Raw images.**
(PDF)

## Acknowledgments

We thank Joel Morgan, Takeshi Ito, and Nicole Butler for the critical reading of the manuscript and many suggestions. We thank Catherine Royer for kindly allowing us to use her microscope. We thank Jorge Frias-Lopez for his help with the RNAseq analysis. We thank the Microbiology Core Facility at CBIS, RPI.

## Author Contributions

**Conceptualization:** Teri N. Hreha, Blanca Barquera.

**Data curation:** Teri N. Hreha, Sara Foreman, Ana Duran-Pinedo, Andrew R. Morris, Patricia Diaz-Rodriguez, J. Andrew Jones, Kristina Ferrara, Anais Bourges, Lauren Rodriguez, Alan R. Hauser, Blanca Barquera.

**Formal analysis:** Teri N. Hreha, Sara Foreman, Blanca Barquera.

**Funding acquisition:** Blanca Barquera.

**Investigation:** Teri N. Hreha.

**Project administration:** Blanca Barquera.

**Resources:** Blanca Barquera.

**Supervision:** Mattheos A. G. Koffas, Mariah Hahn, Blanca Barquera.

**Writing – original draft:** Teri N. Hreha, Sara Foreman, Blanca Barquera.

**Writing – review & editing:** Alan R. Hauser, Blanca Barquera.

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
