## [Decision Letter · Decision Letter 0]

12 Nov 2020

PONE-D-20-25026

The three NADH dehydrogenases of Pseudomonas aeruginosa: their roles in energy metabolism and links to virulence

PLOS ONE

Dear Dr. Barquera,

Thank you for submitting your manuscript to PLOS ONE. After careful consideration, we feel that it has merit but does not fully meet PLOS ONE’s publication criteria as it currently stands. Therefore, we invite you to submit a revised version of the manuscript that addresses the points raised during the review process.

The work is clear, solid and of interest. A concerned was raised by Reviewer #1 about your mice infection data as the results of just one experiment are shown in Fig.8A, although the experiment was repeated twice. Please address this concern, provide the requested details about the mutant strains and revise the manuscript as constructively advised by both reviewers.

We look forward to receiving your revised manuscript.

Kind regards,

Alessandro Giuffrè

Academic Editor

PLOS ONE

Journal Requirements:

Reviewers' comments:

Reviewer's Responses to Questions

**Comments to the Author**

1. Is the manuscript technically sound, and do the data support the conclusions?

Reviewer #1: Partly

Reviewer #2: Yes

2. Has the statistical analysis been performed appropriately and rigorously? 

Reviewer #1: Yes

Reviewer #2: Yes

3. Have the authors made all data underlying the findings in their manuscript fully available?

Reviewer #1: Yes

Reviewer #2: Yes

4. Is the manuscript presented in an intelligible fashion and written in standard English?

Reviewer #1: Yes

Reviewer #2: Yes

5. Review Comments to the Author

Reviewer #1: This manuscript describes the phenotypes of deletion strains for NDH-2, NQR and NUO in P. aeruginosa. The manuscript is well written and clearly describes differences between the reported results and existing literature. The use of LB and SCFM for experimentation, as well as stationary and log phase is helpful. The authors have used PCR to confirm the identities of the transposon strains used.

The finding that MexEF-OprN is differentially regulated in the �Nqr strain is interesting.

The finding that

Major Comments

1. Please show in the manuscript (or provide to the reviewers) the data for the other replicate of the mouse experiment. Demonstration of only one replicate is not adequate for evaluation.

Minor Comments

• Line 63, would briefly mention the ability of P. aeruginosa to anaerobically respire. This adds even more branches to the respiratory chain (and is an acceptable limitation to this data set).

• Format organism names in the bibliography uniformly.

• Table 1: Refer to insertion deletion mutants using ::Tn or something similar to differentiate the from the chromosomal deletions (see https://www.gs.washington.edu/labs/manoil/libraryindex.htm). Are the double mutants done in the ::Tn backgrounds? If so, was the abx cassette from a Tn left in place? For the purposes of the body of the manuscript, the chosen nomenclature is adequate because it improves clarity of the writing.

• For growth curves, please present data as log-OD (Figures 1, 2, S1, etc).

• Table 3: Doublecheck the > and < signs in the ��ndh column

• Line 541 and 542: comment still present in document from Word.

Reviewer #2: Hreha et al. characterize the function of the three NADH dehydrogenases in P. aeruginosa. They determine that NQR has the greatest dehydrogenase activity and that its absence increases the cytotoxin pyocyanin, macrophage death, and mouse mortality. Their results suggest a model in which NADH dehydrogenase activity through NQR negatively regulates pyocyanin and virulence. Overall, the topic is very interesting and the results provide further evidence for a causal link between NADH metabolism and pathogenesis. My only comments are requests for further clarity on the issues below.

Minor issues:

1. Line 308: ref 29 does not appear to be an appropriate reference for SCFM medium.

2. Fig. 7. The y-axis appears to be the fold-change in macrophage death compared to the bacteria-free control. However, the extent of death in bacteria-free control should be the minimal and the death in the other groups is therefore difficult to interpret. I suggest the authors plot something more intuitive such as % cell death or % viability instead.

3. Section starting at line 656: Title mentions biofilm formation but this is not discussed in the paragraph that follows.

4. Line 229: “4V%” looks like a typo.

5. Lines 446 & 447: “propodium” should be “propidium”

6. Line 473: There are no stars in Fig. 5

6. PLOS authors have the option to publish the peer review history of their article (what does this mean?). If published, this will include your full peer review and any attached files.

Reviewer #1: No

Reviewer #2: No

---

## [Author Response · Author response to Decision Letter 0]

2 Dec 2020

Reviewer #1: This manuscript describes the phenotypes of deletion strains for NDH-2, NQR and NUO in P. aeruginosa. The manuscript is well written and clearly describes differences between the reported results and existing literature. The use of LB and SCFM for experimentation, as well as stationary and log phase is helpful. The authors have used PCR to confirm the identities of the transposon strains used.

The finding that MexEF-OprN is differentially regulated in the DNqr strain is interesting.

The finding that

Major Comments

1. Please show in the manuscript (or provide to the reviewers) the data for the other replicate of the mouse experiment. Demonstration of only one replicate is not adequate for evaluation.

We have included data for the replica of the mouse experiments (Figure 8). These data are included in S7 Figure. 

Minor Comments

• Line 63, would briefly mention the ability of P. aeruginosa to anaerobically respire. This adds even more branches to the respiratory chain (and is an acceptable limitation to this data set).

We have modified the text to include the referee’s comment.

• Format organism names in the bibliography uniformly.

This was corrected.

• Table 1: Refer to insertion deletion mutants using ::Tn or something similar to differentiate the from the chromosomal deletions (see https://www.gs.washington.edu/labs/manoil/libraryindex.htm). Are the double mutants done in the ::Tn backgrounds? If so, was the abx cassette from a Tn left in place? For the purposes of the body of the manuscript, the chosen nomenclature is adequate because it impr9ves clarify of the writing.

We have now included a clarification of the nomenclature used for the mutants (Line 11.) The transposon mutants were also described in Table 1 using the “ ::Tn nomenclature”. 

• For growth curves, please present data as log-OD (Figures 1, 2, S1, etc).

We have included new Figures 1 and 2 presenting the growth curves in log scale. The original figures are included as S2 and S3 Figures.

• Table 3: Doublecheck the < and > signs in the Dndh column

This was corrected.

• Line 541 and 542: comment still present in document from Word.

This was corrected.

Reviewer #2: Hreha et al. characterize the function of the three NADH dehydrogenases in P. aeruginosa. They determine that NQR has the greatest dehydrogenase activity and that its absence increases the cytotoxin pyocyanin, macrophage death, and mouse mortality. Their results suggest a model in which NADH dehydrogenase activity through NQR negatively regulates pyocyanin and virulence. Overall, the topic is very interesting and the results provide further evidence for a causal link between NADH metabolism and pathogenesis. My only comments are requests for further clarity on the issues below.

Minor issues:

1. Line 308: ref 29 does not appear to be an appropriate reference for SCFM medium

This was corrected.

2. Fig. 7. The y-axis appears to be the fold-change in macrophage death compared to the bacteria-free control. However, the extent of death in bacteria-free control should be the minimal and the death in the other groups is therefore difficult to interpret. I suggest the authors plot something more intuitive such as % cell death or % viability instead.

We have included a new Figure 7. This figure shows the % of cell death.

3. Section starting at line 656: Title mentions biofilm formation but this is not discussed in the paragraph that follows.

The title has been modified.

4. Line 229: “4V%” looks like a typo.

This was corrected.

5. Lines 446 & 447: “propodium” should be “propidium”

This was corrected.

6. Line 473: There are no stars in Fig. 5.

This was corrected.

---

## [Editor Report · Decision Letter 1]

4 Dec 2020

The three NADH dehydrogenases of Pseudomonas aeruginosa: their roles in energy metabolism and links to virulence

PONE-D-20-25026R1

Dear Dr. Barquera,

We’re pleased to inform you that your manuscript has been judged scientifically suitable for publication and will be formally accepted for publication once it meets all outstanding technical requirements.

Kind regards,

Alessandro Giuffrè

Academic Editor

PLOS ONE
---

## [Editor Report · Acceptance letter]

9 Dec 2020

PONE-D-20-25026R1 

The three NADH dehydrogenases of *Pseudomonas aeruginosa*: their roles in energy metabolism and links to virulence 

Dear Dr. Barquera:

I'm pleased to inform you that your manuscript has been deemed suitable for publication in PLOS ONE. Congratulations! Your manuscript is now with our production department. 

Kind regards, 

on behalf of

Dr Alessandro Giuffrè 

Academic Editor

PLOS ONE